# Preparation and Characterizations of Intrinsically Black Polyesterimide Films with Good Thermal Endurance at Elevated Temperatures for Potential Two-Layer Flexible Copper Clad Laminate Applications

**DOI:** 10.3390/polym17030304

**Published:** 2025-01-23

**Authors:** Shujun Han, Xi Ren, Duanyi Li, Zhenyang Song, Changxu Yang, Zhenzhong Wang, Jingang Liu

**Affiliations:** Engineering Research Center of Ministry of Education for Geological Carbon Storage and Low Carbon Utilization of Resources, School of Materials Science and Technology, China University of Geosciences, Beijing 100083, China; 2103230036@email.cugb.edu.cn (S.H.); renxi@email.cugb.edu.cn (X.R.); 2103240041@email.cugb.edu.cn (D.L.); 1003210426@email.cugb.edu.cn (Z.S.); 2103220040@email.cugb.edu.cn (C.Y.); 3003230013@email.cugb.edu.cn (Z.W.)

**Keywords:** polyimide, flexible copper clad laminate, blackness, optical properties, thermal properties

## Abstract

Polymer films with combined properties of good thermoplasticity, good electrical properties, and good thermal endurance are highly required for two-layer flexible copper clad laminate (FCCL) applications. Meanwhile, the black appearance is also required for specific FCCL applications. Therefore, in the present work, a series of ester-linked polyimide (PEsI) films were designed and developed via the copolymerization chemistry of an ester-containing dianhydride of biphenyl dibenzoate-3,3′,4,4′-tetracarboxylic acid dianhydride (BPTME), a rigid-rod dianhydride of 3,3′,4,4′-biphenyltetracarboxylic acid dianhydride (BPDA), an ester-bridged diamine of 2-(4-aminobenzoate)-5-aminobiphenyl (ABABP), and a functional diamine of 4,4′-iminodianiline (NDA). The molar proportion of the BPTME/BPDA was fixed to be 20:80 and that of ABABP/NDA increased from 50:50 for PEsI-1 to 0:100 for PEsI-VI. The afforded PEsI films showed obviously enhanced blackness with the increasing molar ratio of NDA in the polymers. The PEsI-VI film exhibited the optical transmittance values of 0 and 27.4% at the wavelength of 500 nm (T_500_) and 760 nm (T_760_), respectively. The values were apparently lower than those of the standard PI-ref produced from common pyromellitic dianhydride (PMDA) and 4,4′-oxydianiline (ODA) (T_500_ = 63.2%; T_760_ = 86.3%). Meanwhile, the PEsI-V film showed good blackness with the CIE Lab optical parameters of 1.83 for L*, 11.46 for a*, 3.13 for b*, and 0 for haze. The PEsI samples exhibited good thermoplasticity and the storage and loss modulus of the films rapidly decreased around the glass transition temperatures (T_g_) in the dynamic mechanical analysis (DMA) tests. The PEsI samples revealed the T_g_ values from 247.2 °C to 286.1 °C in the differential scanning calorimetry (DSC) measurements. The PEsI samples exhibited the linear coefficients of thermal expansion (CTE) of (27.1~33.4) × 10^−6^/K from 50 to 250 °C, which was comparable to that of the PI-ref sample (CTE = 29.5 × 10^−6^/K), however, a bit higher than that of the copper foil (CTE = 17.0 × 10^−6^/K).

## 1. Introduction

Flexible copper clad laminates (FCCL) represent a key component achieving the miniaturization and thinning of electronic devices [1]. Generally, FCCLs consist of electrically insulating flexible polymeric substrates and electrically conductive copper conductors. Due to the severe demands on the efficiency, performance, reliability, and the manufacturing procedures of electronic devices, the polymer substrates for FCCLs are usually required to possess good processability and excellent thermal, mechanical, and dielectric properties [2,3,4]. One of the most promising candidates for the polymer substrates used in the fabrication of FCCLs is polyimide (PI) and the related derivatives due to the excellent comprehensive properties of the polymers [5,6,7]. In practice, PIs can be used as the core layers or adhesives for FCCLs depending on the specific construction of the device structure. As shown in Figure 1, there are mainly two structures for the practical FCCLs, including the three-layer one (Figure 1a) and the two-layer form (Figure 1b). For the former, the PI film was used as the core layer, on which the epoxy or acrylic resin was applied as the adhesive to attach the PI and the copper foil. For such applications, the PI core layers are usually characterized by the excellent thermal resistance, high tensile strength, high modulus, and low linear coefficients of thermal expansion (CTE). However, the use of epoxy and acrylic adhesives, on one hand, increases the thickness of the afforded FCCLs and, on the other hand, apparently decreases the thermal endurance class and deteriorates the dielectric features of the FCCL components at the same time [8,9,10,11]. For the latter form, the additional adhesives are eliminated and, as an alternative, a type of thermo-formable PI, usually thermoplastic PIs (TPI), act as the role of adhesives for the attachment of the PIs and copper foil [12]. For such structure, three processing procedures, including the lamination type, the casting type [13], and the physical vapor deposition (PVD) type [14] are often used. Among the procedures, the lamination and casting ones are the most common pathways because of the high production efficiency and relatively low cost. For the lamination type, the bond ply consisting of the sandwich structure of TPI/PI/TPI is usually used to be laminated with the copper foil by a hot-pressing procedure [15]. For such application, the TPI layers are required to have good adhesion and matched CTE values to both of the PI core layer (CTE: 10.0 × 10^−6^/K~30.0 × 10^−6^/K) and the copper foil (CTE: 17.0 × 10^−6^/K), good thermoplasticity, low water uptakes, good dielectric features (low dielectric constant and low dissipation factor), and suitable glass transition temperatures (T_g_: 250~300 °C) so as to achieve high reliability for the fabricated FCCLs [16].

Wang, et al. reported the phosphinated poly(imide siloxane) films for potential two-layer FCCL applications [17]. Although the developed PI films exhibited good thermoplasticity, enhanced adhesion strength to copper foil, and reduced dielectric constants, the films exhibited high CTE values over 50 × 10^−6^/K. Du, et al. reported TPIs derived from the aromatic diamine containing carbazole and short flexible linkages in the molecular chain [18]. The afforded TPI film showed good dimensional stability, with the CTE values in the range of (38~48) × 10^−6^/K. However, the TPI films exhibited over-high T_g_ values over 348 °C, which might deteriorate the hot-pressing procedure for the two-layer FCCLs. In addition, the derived TPI films showed slightly higher dielectric dissipation factors (D_f_). Cao, et al. reported adhesive-free double-sided FCCLs with outstanding adhesion strength [19]. The TPI was derived from the copolymerization of asymmetrical biphenyl dianhydride (aBPDA), the symmetrical biphenyl dianhydride (sBPDA), and ether-containing diamines. The afforded TPI films exhibited appropriate T_g_ values around 258~271 °C and could be laminated with copper foil under the molding temperature of 360 °C with a pressing pressure of 15 MPa for 60 s. The derived two-layer FCCL showed a peel strength of 1.22 N/mm. However, the TPI films showed slightly high D_f_ values over 0.008. Recently, Tan and colleagues reported black TPI films for two-layer FCCL applications [20,21]. The developed PI films showed intrinsic blackness and enhanced peeling strength to copper foil because of the introduction of ketone and ether units. The intrinsic black appearance is valuable for the practical uses of the PI films for FCCLs in view of the intellectual property protection function and the aesthetic consideration [22,23,24]. However, the high CTE and poor thermoplasticity might limit the uses of the black PI films.

In summary, the high CTE and high D_f_ values of the currently developed TPIs greatly prohibited the research and development of the two-layer FCCLs. New TPIs with desirable high-temperature dimensional stability, low dielectric loss, high dielectric strength, and, in many cases, intrinsically black appearance are highly desired. As for the high-frequency dielectric loss, He and colleagues revealed the high-frequency dielectric loss of ester-containing PI or polyesterimide (PEsI) films based on the simulation and experiment validation [25]. The results indicated that the introduction of ester substituents in the PI skeletons could effectively reduce the dielectric loss of the polymers. Meanwhile, the high-frequency dielectric loss of the PEsI films decreased with the increased ester group numbers. As for the CTE of the PI films, the pioneering research work performed in Hasegawa’s group has proven the great efficiency for decreasing the CTE of the PI samples by introduction of the ester linkages in the backbones [26,27,28]. Thus, PEsI films might be a good candidate as the TPIs for two-layer FCCLs. In the present work, the thermoplastic PEsI films were designed and developed for potential applications for two-layer FCCL fabrications.

For the research and development of the TPIs that can meet the property requirements mentioned above, novel PEsI samples were prepared via the polymerization of several functional monomers. The target PEsI films were expected to possess the desired properties for two-layer FCCLs, including good thermal stability, good high-temperature dimensional stability, low dielectric constants and dissipation factor, and so on. It is quite difficult to achieve all the properties into one single homo-polymerized PEsI film. Thus, several monomers with desired function had to be used in the design and preparation of the expected PEsI films. The monomers included the ester- and biphenyl-containing dianhydride of biphenyl dibenzoate-3,3′,4,4′-tetracarboxylic acid dianhydride (BPTME), the rigid 3,3′,4,4′-biphenyltetracarboxylic acid dianhydride (BPDA), and the ester-linked and phenyl-substituted diamine of 2-(4-aminobenzoate)-5- aminobiphenyl (ABABP). Meanwhile, an aromatic diamine of 4,4′-iminodianiline (NDA) was also used to afford the derived PEsI films intrinsically black appearance. The structure–property relationships of the afforded PEsI films were systematically investigated.

## 2. Materials and Methods

### 2.1. Materials

Two commercially available BPDA and BPTME dianhydrides with a purity over 99.0% were bought from TCI Co., Ltd. (Tokyo, Japan) and dried at 180 °C at −0.095 MPa overnight before use. Two aromatic diamines with a limited commercial pathway, including ABABP and NDA, were prepared in house according to the well-established routes and purified by recrystallization and decolored to achieve purities higher than 99.0% [29]. The electron-grade polymerization solvent of N-methyl-2-pyrrolidinone (NMP) (water content ≤ 50 ppm; purities ≥ 99.5%) was purchased from InnoChem Sci. Technol. Co., Ltd. (Beijing, China).

### 2.2. Measurements

Poly(amic acid) (PAA): number average (M_n_) and weight average (M_w_) molecular mass, gained from a gel permeation chromatography (GPC) measurement (Shimadzu Co., Ltd., Kyoto, Japan) using NMP and polystyrene as the mobile phase and standard samples, respectively.

PEsI film:

Fourier transform infrared (FTIR) spectra: gained from a Tensor-27 FT-IR spectrometer (Bruker, Ettlingen, Germany).

Ultraviolet-visible (UV-Vis) spectra: gained from the Hitachi U-3210 spectrophotometer (Tokyo, Japan) at room temperature.

Wide-angle X-ray diffraction (XRD): gained on a D/max-2500 X-ray diffractometer (Rigaku, Tokyo, Japan). The intramolecular distance (d) was gained by the Bragg’s equation of *n*λ = 2dsinθ, where *n* = 1; λ = 1.5418 Å (X-ray wavelength); and θ stands for the half of the X-ray scattering angle.

CIE Lab color parameters: gained from a color i7 spectrophotometer (X-rite, Grand Rapids, MI, USA). L* (lightness) ranges from 0 (black) to 100 (white). The parameter of a* indicates red (positive)–green (negative) indices, while the b* means yellow (positive)–blue (negative) indices. The whiteness indices (WI) were obtained as the calculation of WI = 100 − [(100 − L*)^2^ + a*^2^ + b*^2^]^1/2^.

Thermogravimetric analysis (TGA): gained from a Q50 thermal analysis system (TA Instruments, New Castle, DE, USA) from room temperature to 750 °C in nitrogen (heating rate: 20 °C/min).

Differential scanning calorimetry (DSC) was carried out on a DSC 200 F3 Maia system (NETZSCH, Selb, Germany) at a heating rate of 10 °C/min in nitrogen.

Dynamic mechanical analysis (DMA): gained from a Q800 thermal analysis system (TA Instruments, New Castle, DE, USA) from room temperature to 350 °C in nitrogen (heating rate: 5 °C/min; frequency: 1 Hz).

Thermo-mechanical analysis (TMA): gained on a TMA402F3 thermal analysis system (NETZSCH, Selb, Germany) from 50 to 300 °C in nitrogen (heating rate: 5 °C/min). The coefficients of linear thermal expansion (CTE) values were recorded from 50 °C to 250 °C.

Tensile properties: obtained on a 3365 universal testing system (Instron, Grove City, PA, USA) with 80 mm × 10 mm × 0.05 mm specimens at a rate of 1.0 mm/min. The tensile strength (T_S_) and tensile modulus (T_M_) were obtained.

Dielectric strength: gained on an IBV-5/20 apparatus (Guilin, China) with a rate of 0.5 kV/s with the sample sizes of 1.5 cm × 1.5 cm × 0.025 mm (length × width × thickness).

Dielectric constant (D_k_) and dielectric dissipation factors (D_f_): gained on a 4294A precise impedance analyzer (Agilent, Santa Clara, CA, USA) (testing temperature: 25 °C; frequency: 10 GHz) using the average results of five samples.

### 2.3. Preparation of PEsI Films

The PEsI samples were prepared based on the formula shown in Table 1 by a standard two-stage polycondensation pathway through the poly(amic acid) (PAA) precursors. PEsI-I could be used as an example to illustrate the preparation steps. First, ABABP (15.2170 g, 0.05 mol) and NDA (9.9625 g, 0.5 mol) were added into a 500 mL flask equipped with an Ika Eurostar 60 Control mechanical stirrer (Staufen, Germany), a cold water bath, and a nitrogen inlet and outlet system containing the ultra-dry NMP (150.0 g) at 5–10 °C. After stirring under nitrogen for 20 min, a pale-purple diamine solution was obtained. Visually, the ABABP diamine showed relatively worse solubility than that of the NDA diamine. Then, BPTME dianhydride (10.6886 g, 0.02 mol) and BPDA dianhydride (23.5376 g, 0.08 mol) were added to the diamine solution within 10 min. Residual NMP (28.0 g) was added into the polymerization system to afford a reaction system with a solid content of 25 wt%. Moderate exothermic phenomena could be observed during stirring at 5–10 °C, indicating that polymerization occurred. The cold bath was removed after 3 h, at which time a viscous suspension formed. The reaction system was stirred at room temperature until the total polymerization time reached 24 h. The deep-color PAA solution was then diluted to be 15 wt% with dry NMP and stirred rapidly to form a homogeneous varnish.

The obtained PAA-I solution was coated onto clean glass substrate and converted to the final PEsI-I film by the thermal imidization procedure of 80 °C/1 h, 150 °C/1 h, 250 °C/1 h, and 300 °C/1 h in a high-temperature oven. The PEsI-I specimen was peeled off the substrate by placing the glass substrate in the hot water bath at 80 °C. The film exhibited good adhesion to the glass and it took several hours to separate from the substrate even being immersed in the hot water. The wet film was further heated in an air-circulating oven at 120 °C for 3 h.

The other PAA and the corresponding PEsI films were prepared with similar procedures like PAA-I and PEsI-I, respectively.

## 3. Results and Discussion

### 3.1. PEsI Preparation

The copolymerized PEsI specimens were prepared according to the two-step procedures shown in Figure 2. The molar ratio of BPTME/BPDA was set to be 20/80 based on our previous study [29]. At this ratio, the derived PEsI films showed the best comprehensive properties as a class of high-performance thermoplastics. For the diamine units, the molar proportions of the NDA, which serves as the function of “colorant” in the derived black PEsI films, increased from 50 mol% for PEsI-I to 100 mol% for PEsI-VI sample. As expected, the color of the PEsI films might gradually deepen with the increase in the NDA in the specimens, but some other properties might deteriorate as a result, which will be discussed in detail later.

First, the PAA precursors were synthesized and the molecular mass values were detected by GPC measurements, whose data and plots are shown in Table 2 and Figure 3, respectively. All the copolymers showed high molecular mass values over 10^4^ g/mol and polydispersity indices (PDI) in the range of 2.00 to 2.63, indicting the good polymerization reactivity of the four monomers and fewer side reactions in the polycondensation. In addition, the molecular mass, either the number average (M_n_) or the weight average (M_w_) ones, of the PAAs gradually increased with the increasing molar contents of NDA in the diamine units (Figure 3). For example, PAA-I showed an M_n_ value of 2.37 × 10^4^ g/mol, which was apparently lower than that of PEsI-VI (M_n_ = 6.70 × 10^4^ g/mol). This might be due to the relatively higher reactivity of NDA compared with ABABP. For the latter, the electron-withdrawing ester linkages and the bulky phenyl substituents in the molecular structures deteriorated the reactivity of the amino groups based on either the electronic effects or the steric hindrance effects. Nevertheless, the current PAA precursors were successfully cast into tough PEsI films. The rigid-rod biphenyl and ester linkages in the dianhydride units and the main-chain ester bonds and the lateral phenyl substituents in the diamine units endowed the PEsI films with good tensile properties, with a tensile strength (T_S_) over 148 MPa and a tensile modulus (T_M_) over 5.1 GPa. The good tensile feature was beneficial for the practical applications of the current PEsI specimens in the two-layer FCCLs.

Figure 4 depicts the FTIR test results of the PEsI specimens, in which the characteristic absorptions of the composition structures in the polymers could be clearly identified. For example, the imide rings revealed the characteristic absorptions at 1776 cm^−1^ (asymmetric carbonyl stretching vibrations), 1725 cm^−1^ (symmetric carbonyl stretching vibrations), 1391 cm^−1^ (C-N stretching vibrations), and 738 cm^−1^ (carbonyl bending vibrations), respectively. The ester carbonyl groups exhibited the characteristic absorptions at 1697 cm^−1^. The secondary amine (-NH-) groups in the NDA units and the unsaturated C-H bonds in the phenyl units showed the characteristic absorptions at 3405 cm^−1^ and 3057 cm^−1^, respectively. At last, the unsaturated -C=C- bonds in phenyl rings showed the characteristic absorptions at 1613 cm^−1^ and 1531 cm^−1^, respectively. The FTIR results indicated the obtaining of the polymers.

Figure 5 presents the XRD plots of the PEsI samples labeled with the scattering angles (2 theta or 2θ) and the corresponding d-spacing values calculated by Bragg’s equation. It could be observed that slightly crystalline regions existed inside the PEsI films, which manifested as the sharp absorption peaks in the plots. The microcrystalline regions were mainly caused by the rigid biphenyl and ester bond groups in the PEsI molecular structure, which was in good agreement with our previous work on the ABABP-derived PIs [29]. At the same time, through careful observation, it could be seen that there were two crystallization peaks in the XRD plots of the PEsI films, which were found in the range of 14~15° and 20~25°, respectively. Correspondingly, the d_1_ and d_2_ values were calculated according to the two individual peaks and are shown in the figure. Basically, with the decreasing contents of ABABP in the molecular structures of PEsI films (from PEsI-I to PEsI-VI), the d-spacing values of the samples showed a downward trend, especially for d_2_. This was mainly due to the ABABP diamine containing a large molar volume of phenyl substituents in the side chain, which significantly increased the d values of the intramolecular chains.

### 3.2. Optical and Dielectric Properties

In many practical applications for two-layer FCCLs, TPIs with “pure blackness” are often desired in view of the underlayer circuit protection and the aesthetic consideration [23]. In practice, it is difficult to achieve a pure blackness for a polymer film, which is usually quantitatively expressed by the optical parameters of T_400~760_ = 0 (average optical transmittance from 400 nm to 760 nm), L* = 0 (CIE lightness), and a* = b* = 0 (CIE optical indices). Intrinsically black polymer films are often more advantageous in FCCL applications than those counterparts achieving blackness via surface coating with black paints or internally combining with black fillers due to the potentially deteriorated electrical and tensile properties of the composites. Monomers with black chromophores, such as the currently used NDA diamine, are often to be adopted to develop intrinsically black PI films. However, the usage of NDA in the research and development of intrinsically black thermoplastic PEsI films has rarely been reported.

Figure 6 depicts the UV-Vis spectra of the PEsI films and the referenced PI-ref film based on pyromellitic dianhydride (PMDA) and 4,4′-oxydianiline (ODA), together with the appearance of the samples. The optical data of the films are tabulated in Table 3. Visually, the PEsI samples all presented black appearance, while the PI-ref sample showed the standard brown–yellow color. Quantitatively, the PEsI films showed a cutoff wavelength (λ_cut_) over 500 nm, which was nearly 100 nm higher than that of the standard PI-ref sample (λ_cut_ = 407 nm). The PEsI samples exhibited the totally opaque feature at the wavelength of 500 nm (T_500_ = 0) and exhibited optical transmittance values that decreased from 54.0% (PEsI-I) to 27.4% (PEsI-VI) at the wavelength of 760 nm with the increasing contents of NDA in the polymers. Comparatively, the PI-ref showed T_500_ and T_760_ values of 63.2% and 86.3%, respectively.

The optical feature of the PEsI films could be interpreted as a result of the enhancement of intermolecular and intramolecular charge transfer (CT) actions in the polymer chains due to the improvement in the electron-donating ability of the diamine units by the electron-rich NDA components [30]. This enhancement of the charge transfer could be deduced from the molecular frontier orbit simulation of the PEsI samples, as shown in Figure 7. The lowest unoccupied molecular orbital (LUMO) energy levels (ε_LUMO_) and the highest occupied molecular orbital (HOMO) energy levels (ε_HOMO_) of several representative PI homopolymers, including PI (BPTME-ABABP, Figure 7a), PI (BPTME-NDA, Figure 7a), PI (BPDA-ABABP, Figure 7b), and PI (BPDA-NDA, Figure 7b) were simulated based on the density functional theory (DFT)/B3LYP methods. Meanwhile, the 6-311G(d) basis set was used with the Gaussian 09 software [31]. The energy level gaps (E_g_) of the above-mentioned polymers were calculated by the equation of E_g_ = ε_LUMO_-ε_HOMO_ and summarized in Figure 7c. It has been well established in the literature that the degree of CT interactions could be roughly estimated by the E_g_ values and a lower E_g_ usually indicated a higher CT degree in the polymer chains [32]. Based on the data shown in Figure 7c, the E_g_ values of the PIs decreased with the order of PI (BPTME-ABABP) > PI (BPDA-ABABP) > PI (BPTME-NDA) > PI (BPDA-NDA). This indicated that the coloration of the PEsI samples mainly came from the contribution of the NDA component instead of ABABP. In addition, one could observe that the ε_HOMO_ values for the NDA-derived polymers (BPTME-NDA and BPDA-NDA) were all higher than those of the ABABP-derived ones (BPTME-ABABP and BPDA-ABABP). This implied a higher electron-donating ability of the NDA components. On the other hand, in the case of the same diamine, the contribution of BPDA dianhydride to the film coloration was higher than that of BPTME. This was mainly due to the presence of highly conjugated biphenyl structures in the BPDA dianhydride.

The structure–coloration relationships of the PEsI films could also be deduced by the CIE Lab color parameter measurements tabulated in Table 3. With the increase in NDA components, the samples exhibited gradually decreased lightness (L*), red–green indices (a*), and yellow–blue indices (b*). This trend indicated the gradually deepened colors of the PEsI films with increasing NDA components. Quantitatively, the whiteness indices (WI) of the PEsI films decreased from 5.26 for PEsI-I to 1.11 for PEsI-VI, which gradually approached pure blackness (L* = a* = b* = 0, WI = 0). Comparatively, the PI-ref (PMDA-ODA) film showed a WI value of 19.24. Apparently, the molecular design of incorporating the NDA components into the PEsI samples to endow the films with intrinsic blackness was feasible and effective.

The dielectric characteristics of the PEsI samples, including the dielectric strength (D_S_), dielectric constant (D_k_), and dielectric dissipation factors (D_f_) are summarized in Table 3. The PEsI sample exhibited the D_S_ values in the range of 156~178 V/μm. The D_S_ values were lower than those of the PEsI films reported previously (D_S_ ≥ 220 V/μm) [29]. This might be attributed to the good electron mobility in the NDA components in the current black PEsI polymers [33]. In addition, the PEsI samples exhibited D_k_ values from 3.38 to 3.85 and D_f_ values from 0.00104 to 0.00996 at 10 GHz, respectively. It could be deduced that the series of black films showed an excellent low-D_f_ feature, which was one of the most important characteristics of the PEsI samples [34]. The D_k_ values of the series of PEsI films were close to that of the standard PI samples.

### 3.3. Thermal Properties

Thermal resistance of the black PEsI films were investigated by various measurements and the thermal data are summarized in Table 4. First, Figure 8 depicts the TGA as well as the corresponding derivative TG (DTG) curves of the PEsI samples. All the samples maintained most of the original weights before 400 °C; after that, obvious decompositions were observed. The 5% weight loss temperatures (T_5%_) were found to be from 488 °C to 503 °C. The dual decomposition phenomena were clearly recorded in the DTG curves. The temperatures corresponding to the first rapidest decomposition (T_max1_) and the second rapidest decomposition (T_max2_) were detected in the range of 510–525 °C and 607–620 °C, respectively. The first decomposition might be attributed to the decomposition of the ester linkages in the polymers, while the second one was mainly due to the decomposition of the residual polymer skeletons. At 750 °C, the polymers left about 60 wt% of their original weights, reflecting the good thermal resistance of the samples.

The glass transition temperatures (T_g_) of the PEsI samples were gained by DSC and DMA measurements and the obtained T_g_ values were termed as “T_g, DSC_” and “T_g, DMA_”, respectively. Figure 9 presents the DSC plots of the PEsI samples. No obvious melting peaks were observed for all the polymers, although slight crystallinity was detected in the XRD plots (Figure 5). Obvious glass transitions were observed in the tests and the T_g, DSC_ values increased from 247.2 °C for PEsI-I to 286.1 °C for PEsI-VI. The moderate levels of T_g, DSC_ values were beneficial for the hot-pressing procedures in the formation of two-layer FCCLs.

Further, Figure 10 presents the DMA test results for the PEsI. It could be deduced from Figure 10a that the currently developed PEsI specimens exhibited a storage modulus (E′) as high as 5.0–6.0 GPa at room temperature. This was in good consistency with the tensile measurements (Table 2). The polymers maintained the initial E’ values up to 250 °C. After this temperature, the E’ of the PEsI specimens dropped rapidly, indicating good thermoplasticity, except PEsI-VI. For PEsI-VI, the E’ value decreased slowly, reflecting the relatively poor thermoplastic feature of the polymer. This means that incorporation of a high-level NDA component was disadvantageous to maintain the thermoplastic behaviors of the PEsI samples. Nevertheless, the other PEsI copolymers exhibited good thermoplasticity and might exhibit good melt-processability in the fabrication of two-layer FCCLs. Clear T_g, DMA_ values were revealed by the tan delta curves depicted in Figure 10b. The T_g, DMA_ values of the PEsI specimens exhibited a similar trend to that of DSC tests. PEsI-I and PEsI-VI showed the lowest and the highest T_g, DMA_ values of 264.9 °C and 325.6 °C, respectively.

The thermal endurance of the developed PEsI samples at elevated temperatures was investigated by TMA tests. Figure 11 presents the TMA curves of the PEsI samples and the CTE data are tabulated in Table 4. Incorporation of NDA components in the diamine units slightly deteriorated the thermal endurance of the polymers. The PEsI sample derived from BPTME, BPDA, and ABABP showed a CTE value of 15.3 × 10^−6^/K [29], while the data increased to (27.1–33.4) × 10^−6^/K for the current polymers. The CTE values were a bit higher than that of the copper foil (~17.0 × 10^−6^/K). This might be due to the flexible imino (–NH–) bridges in the NDA components. Although the thermal endurance of the PEsI samples was slightly deteriorated, they still showed acceptable CTE values for practical two-layer FCCL fabrications.

## 4. Conclusions

Thermoplastic PEsI films with intrinsically black colors were designed and developed via the copolymerization of ester-containing BPTME dianhydride and ABABP diamine, the biphenyl dianhydride, and the black chromophore-containing NDA diamine. Incorporation of the NDA components efficiently enhanced the CT actions in the polymer segments, thus obviously decreasing the optical clearance of the PEsI samples in the visible light range and increasing the T_g_ values of the afforded polymers. However, the thermal endurance and dielectric strength of the developed PEsI samples were deteriorated to some extent. PEsI-III showed the best combined properties, including a T_g,DSC_ of 255.4 °C, a CTE of 27.8 × 10^−6^/K, a D_k_ of 3.67, a D_f_ of 0.00781 at 10 GHz, a T_S_ of 154.9 MPa, a T_M_ of 5.59 GPa, and a WI of 3.94. The technological properties of the PEsI-III film in the fabrication of two-layer FCCLs, such as the thermo-processability, adhesion to copper foil, and so on, will be evaluated in future work.

## Figures and Tables

**Figure 1 polymers-17-00304-f001:**
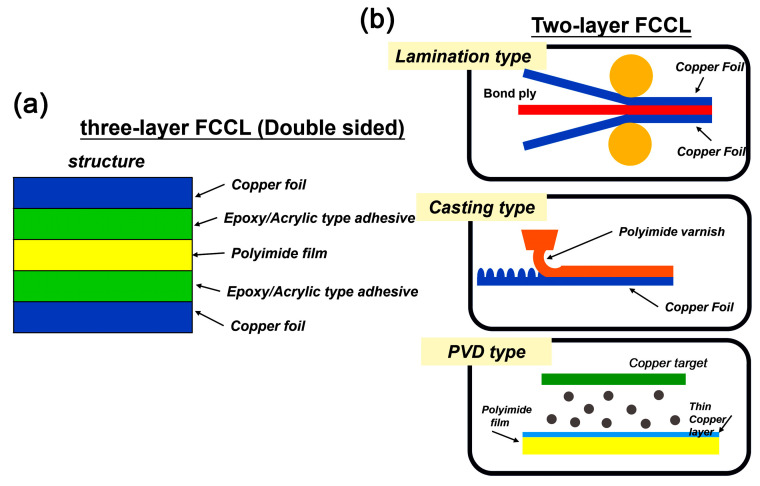
Comparison of the typical structures of the three-layer (**a**) and two-layer FCCLs (**b**).

**Figure 2 polymers-17-00304-f002:**
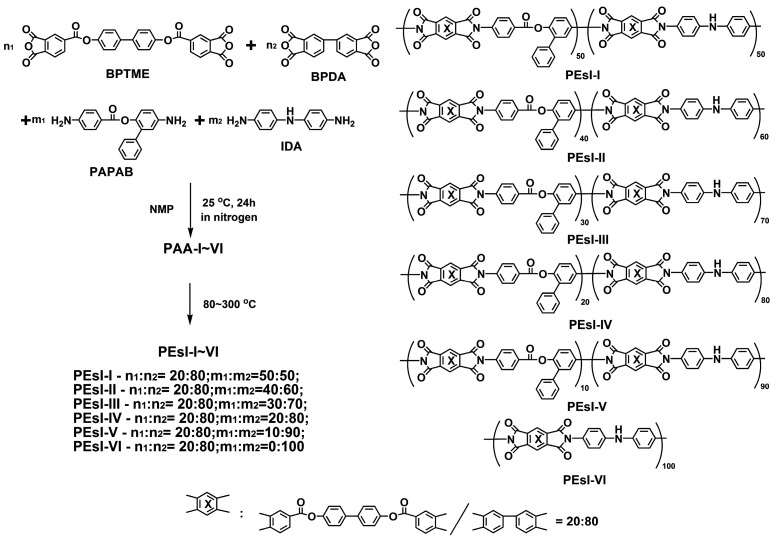
Preparation route of the PEsI specimens.

**Figure 3 polymers-17-00304-f003:**
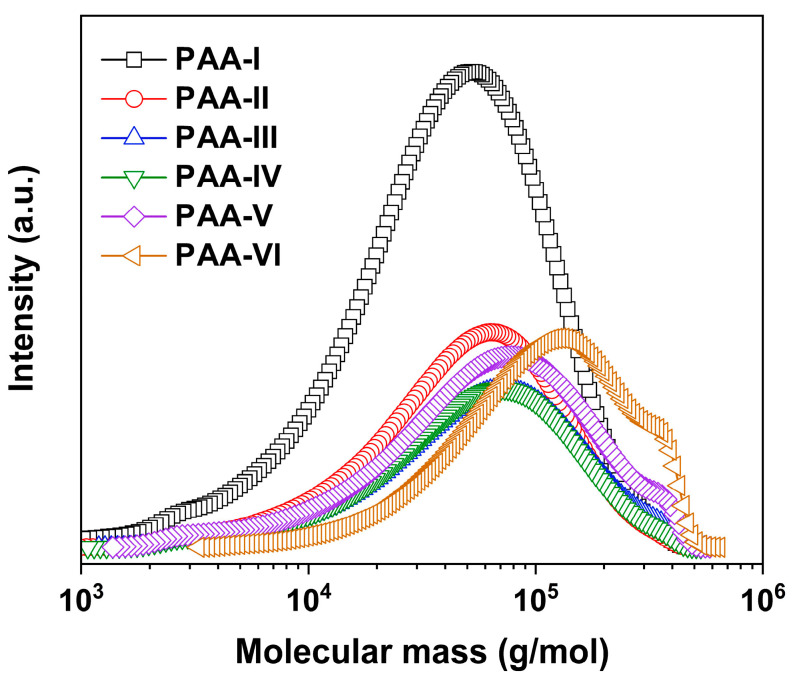
GPC plots of PAA precursors.

**Figure 4 polymers-17-00304-f004:**
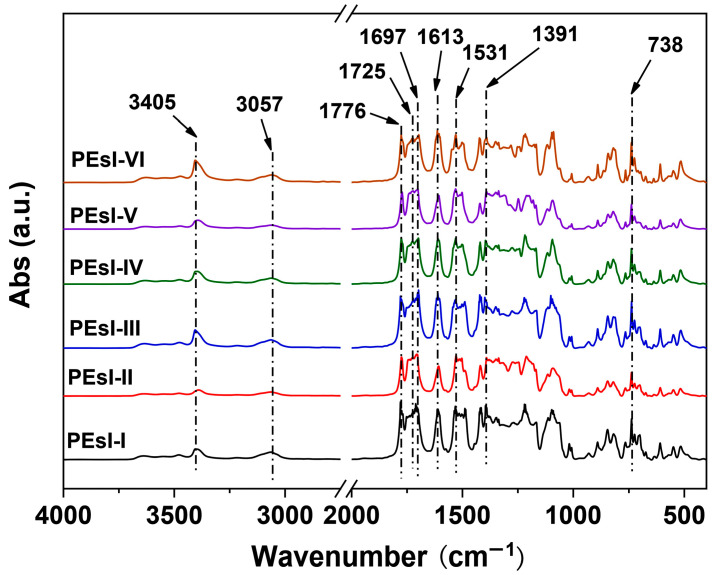
FTIR spectra of the PEsI specimens.

**Figure 5 polymers-17-00304-f005:**
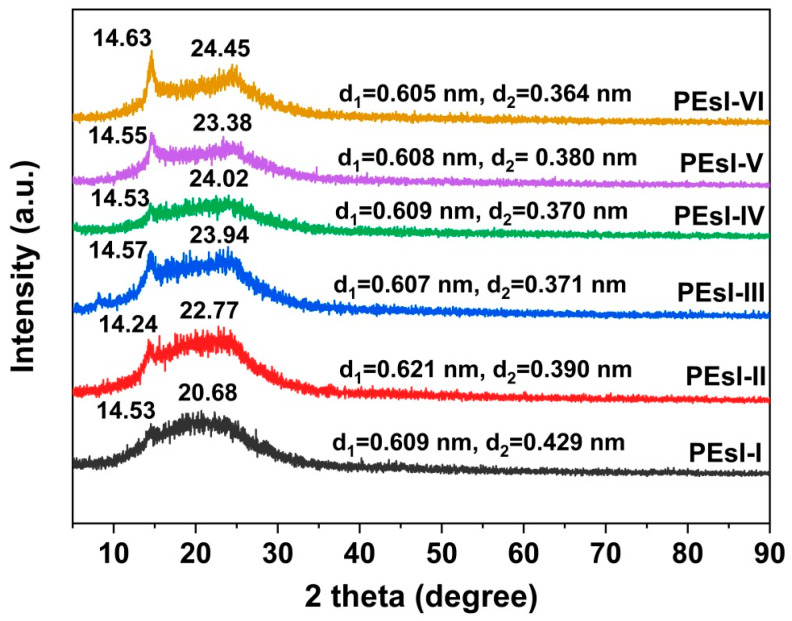
XRD pattern of the PEsI specimens.

**Figure 6 polymers-17-00304-f006:**
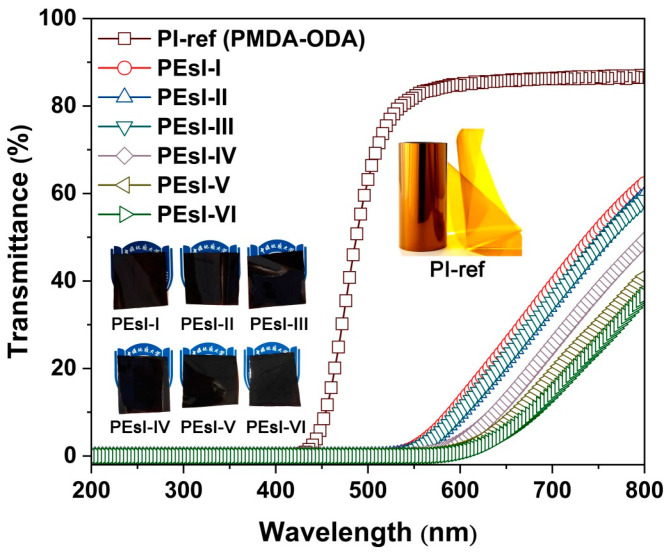
UV-Vis spectra and the appearance of the PEsI and PI-ref specimens.

**Figure 7 polymers-17-00304-f007:**
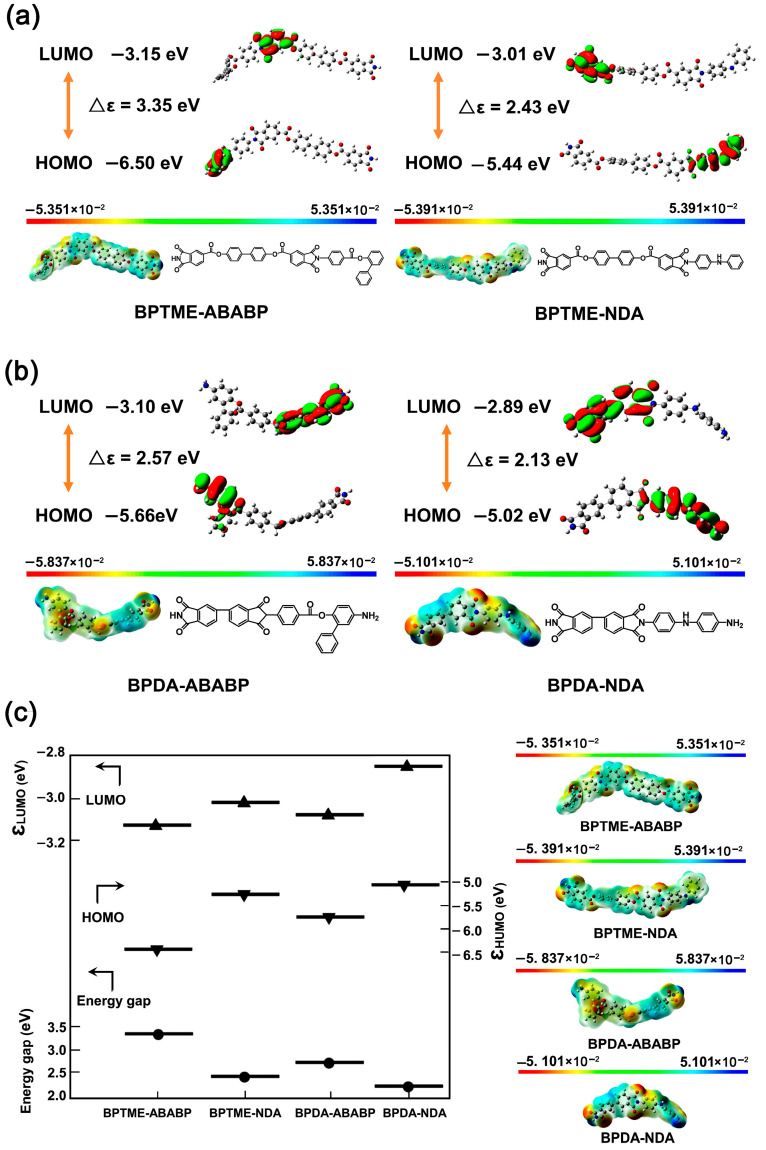
Frontier orbit simulation results for the PEsI specimens. (**a**) BPTME-derived PEsIs; (**b**) BPDA-derived PEsIs; (**c**) orbit energy and energy gap of PEsIs.

**Figure 8 polymers-17-00304-f008:**
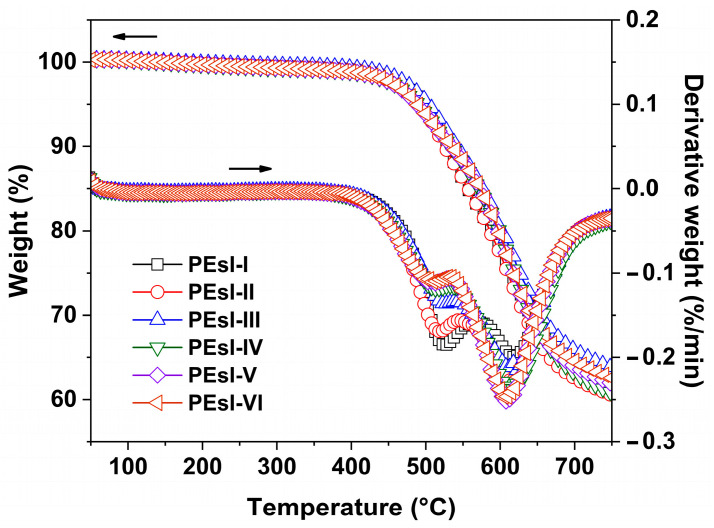
TGA and DTG plots of PEsI specimens.

**Figure 9 polymers-17-00304-f009:**
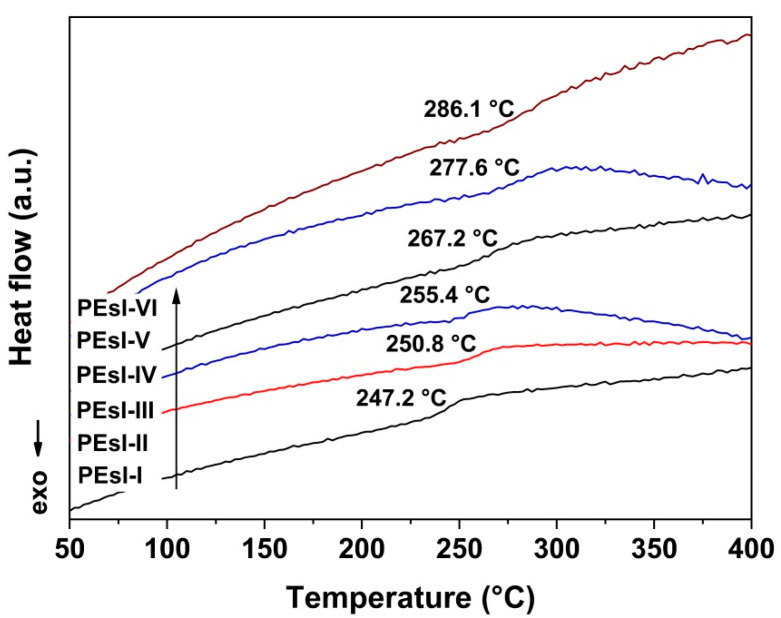
DSC curves of PEsI specimens.

**Figure 10 polymers-17-00304-f010:**
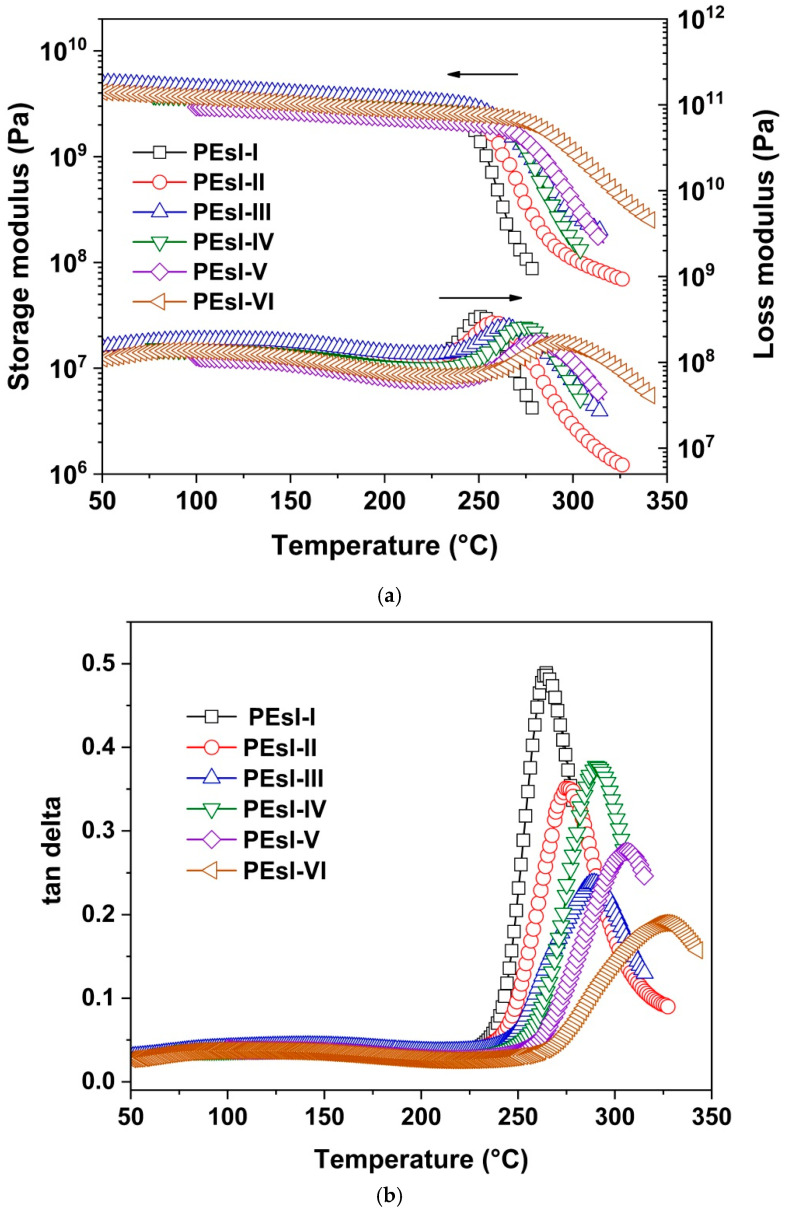
Storage and loss modulus (**a**) and tan delta (**b**) of PEsI samples in DMA tests.

**Figure 11 polymers-17-00304-f011:**
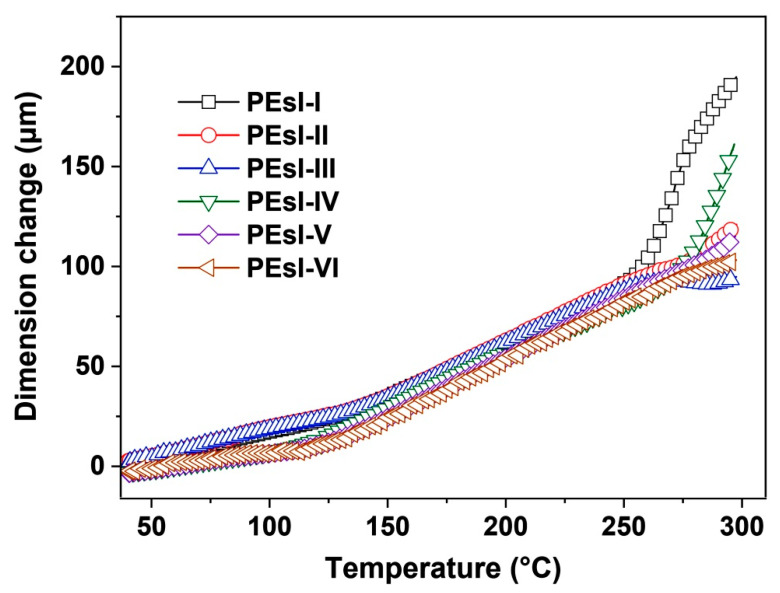
TMA curves of PEsI samples.

**Table 1 polymers-17-00304-t001:** Formula for the synthesis of PAA-I~PAA-VI specimens.

Samples	BPTME (g, mol)	BPDA (g, mol)	ABABP (g, mol)	NDA (g, mol)	NMP (g)
PAA-I	10.6886, 0.02	23.5376, 0.08	15.2170, 0.05	9.9625, 0.05	178
PAA-II	10.6886, 0.02	23.5376, 0.08	12.1736, 0.04	11.9550, 0.06	175
PAA-III	10.6886, 0.02	23.5376, 0.08	9.1302, 0.03	13.9475, 0.07	172
PAA-IV	10.6886, 0.02	23.5376, 0.08	6.0868, 0.02	15.9400, 0.08	169
PAA-V	10.6886, 0.02	23.5376, 0.08	3.0434, 0.01	17.9325, 0.09	166
PAA-VI	10.6886, 0.02	23.5376, 0.08	0, 0	19.9250, 0.10	162

**Table 2 polymers-17-00304-t002:** Molecular mass of PAAs and the qualities of the PEsI specimens.

Samples	M_n_ ^a^ (×10^4^ g/mol)	M_w_ ^a^ (×10^4^ g/mol)	PDI ^a^	Appearance	T_S_ ^b^ (MPa)	T_M_ ^b^ (GPa)
PAA-I	2.37	6.13	2.58	translucent & black	148.4	5.52
PAA-II	3.07	7.07	2.30	opaque & black	149.3	5.67
PAA-III	3.23	7.79	2.41	opaque & black	154.9	5.59
PAA-IV	3.44	8.29	2.41	opaque & black	158.0	5.81
PAA-V	3.47	9.15	2.63	opaque & black	151.8	5.12
PAA-VI	6.70	13.38	2.00	opaque & black	161.1	5.44

^a^ M_n_, M_w_, see Section 2.2; PDI: polydispersity index, M_w_/M_n_; ^b^ T_S_, T_M_, see Section 2.2.

**Table 3 polymers-17-00304-t003:** Optical and dielectric data of the PEsI and PI-ref specimens.

Samples	λ_cut_ ^a^ (nm)	T_500_ ^a^ (%)	T_760_ ^a^ (%)	L* ^a^	a* ^a^	b* ^a^	WI	Haze (%)	D_S_ ^b^ (V/μm)	D_K_ ^b^	D_f_ ^b^
PEsI-I	505	0	54.0	25.41	38.68	43.78	5.26	10.5	156	3.85	0.00996
PEsI-II	504	0	50.9	23.74	38.06	40.91	5.46	0	167	3.69	0.00104
PEsI-III	529	0	50.3	13.15	34.22	22.65	3.94	0	156	3.67	0.00781
PEsI-IV	500	0	40.3	11.21	34.20	19.32	2.91	0	178	3.76	0.00826
PEsI-V	536	0	32.1	5.89	28.29	10.14	1.21	0	158	3.38	0.00645
PEsI-VI	547	0	27.4	1.83	11.46	3.13	1.11	0	169	3.68	0.00889
PI-ref	407	63.2	86.3	88.65	−9.35	79.41	19.24	0.68	ND ^c^	ND	ND

^a^ λ_cut_: cutoff wavelength; T_500_, T_760_: transmittance at the wavelength of 500 nm and 760 nm, respectively; L*, a*, b*, see Section 2.2; WI: whiteness index. ^b^ D_S_, D_k_, D_f_, see Section 2.2; ^c^ not detected.

**Table 4 polymers-17-00304-t004:** Thermal data of the PEsI samples.

PI	T_g, DSC_ ^a^ (°C)	T_g, DMA_ ^a^ (°C)	T_5%_ ^a^ (°C)	T_max1_ ^a^ (°C)	T_max2_ ^a^ (°C)	R_w750_ ^a^ (%)	CTE ^a^ (×10^−6^/K)
PEsI-I	247.2	264.9	498	525	618	62.7	27.1
PEsI-II	250.8	275.8	493	520	620	60.5	27.9
PEsI-III	255.4	288.8	503	525	610	63.8	27.8
PEsI-IV	267.2	290.6	490	515	618	60.8	33.4
PEsI-V	277.6	306.0	488	510	608	61.9	32.0
PEsI-VI	286.1	325.6	490	510	607	62.6	31.1

^a^ T_g, DSC_: glass transition temperatures determined by the DSC tests; T_g, DMA_: glass transition temperatures determined by the DMA (peak temperatures of tan delta) measurements, T_5%_: temperatures at 5% weight loss; T_max1_, T_max2_: temperatures corresponding to the first and second thermal decomposition, respectively; R_w750_: residual weight percent at 750 °C; CTE: linear coefficients of thermal expansion from 50 °C to 250 °C.

## Data Availability

Data are contained within the article.

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
