# Peer review of "Preparation and Characterizations of Intrinsically Black Polyesterimide Films with Good Thermal Endurance at Elevated Temperatures for Potential Two-Layer Flexible Copper Clad Laminate Applications"

_polymers, 2025, doi:10.3390/polym17030304_

Round 1
Reviewer 1 Report
Comments and Suggestions for Authors
1. please explain the importance of black appearance of the films
2. what make using the polymerization of several monomers necessary to create the PEsI films compare to other available materials?
3. the ending of the sentence in line 112 is missing
4. provide the reasoning to the chosen weights of the chemicals for synthesis, how the process was developed?
5. please explain the approach taken to create the frontier orbit simulation
6. lines 364-366 aren't scientifically sound, as no polymers maintains their integrity above 500C and can't be mentioned as example of thermal resistance at 750
7. what is the reason behind the difference in Tg obtained by DSC and DMA techniques - which one we should rely on?
8. please proof read the text again to eliminate the word repetitions as 'increase' in line 380, extra space in 388, typo TMD instead of TMA in line 405, etc.
9. this is the first time I see the term 'thermal durance' - how did you come across it?
10. how many times each provided tests were repeated, what is the reproducibility, the percentage of experimental error?
Author Response
- Question: please explain the importance of black appearance of the films
Answer: Thanks a lot. The importance of the black appearance of the films was added in our revised manuscript as follows.
“…The intrinsic black appearance is valuable for the practical uses of the PI films for FCCLs in view of the intellectual property protection function and the aesthetic consideration [22-24]. ”.
- Question: What make using the polymerization of several monomers necessary to create the PEsI films compare to other available materials?
Answer: Thanks a lot. The target PEsI films were expected to possess the desired properties for 2-layer FCCLs, including god thermal stability, good high-temperature dimensional stability, low dielectric constants and dissipation factor and so on. It is quite difficult to achieve all the properties into one single homo-polymerized PEsI film. Thus, several monomers with desired function had to be used in the design and preparation of the expected PEsI films.
The corresponding discussion was added in our revised manuscript.
- Question:The ending of the sentence in line 112 is missing.
Answer: Thanks a lot. The sentence was supplemented in our revised manuscript as follows.
“…In the present work, the thermoplastic PEsI films were designed and developed for the potential applications for two-layer FCCL fabrications.”.
- Question: Provide the reasoning to the chosen weights of the chemicals for synthesis, how the process was developed?
Answer: The weights of the chemicals were chosen according to the structural design of the copolymerized PEsI films. The molar ratio of the two dianhydrides was set to be 20:80 and the one for ABABP/NDA (m1/m2) changed from 50:50 (PEsI-1) to 0:100 (PEsI-VI). The copolymerization was carried out according to the process presented in section 2.3 in the manuscript.
- Question:Please explain the approach taken to create the frontier orbit simulation.
Answer: Thanks a lot. The frontier orbit simulation was taken based on the density functional theory (DFT)/B3LYP methods. B3LYP is by far the most popular density functional in chemistry (Chem Commun, 2010, 46: 3057-3070). Meanwhile, the 6-311G(d) basis set was used. This combination was often used to simulate the frontier orbit of polyimides and the derivatives. All the simulation was conducted with Gaussian 09 software.
- Question:Lines 364-366 aren't scientifically sound, as no polymers maintains their integrity above 500C and can't be mentioned as example of thermal resistance at 750.
Answer: Thanks a lot. The TGA was conducted from 50 oC to 750 oC in nitrogen environments. Indeed, in air environments, all the polymers might decompose wholly at 750 oC. However, in inert environments, more than half of their original weights were maintained at such high temperature. This is one of the most important features for the thermal-resistant PI polymers.
- Question:What is the reason behind the difference in Tg obtained by DSC and DMA techniques - which one we should rely on?
Answer: Thanks a lot. The difference in Tg obtained by DSC and DMA techniques is mainly relied on the different test conditions for the polymers. DSC is a kind of static thermal analysis technique, which determined the Tg values of the polymers via the heat flux changes as a function of temperature (Chemical Physics Letters, 2007, 440: 372–377). Contrarily, DMA is a kind of dynamic thermal analysis technique, which determined the Tg values of the polymers via the modulus changes as a function of temperature. Different polymers might exhibit different sensitivities to DSC and DMA tests. Some polymers cannot reveal clear glass transition behaviors in the DSC tests. For PIs, generally, if the Tg values could be clearly obtained by DSC tests, then they could be relied on. However, if the Tg values could not be clearly obtained by DSC tests, the data based on DMA tests have to be relied on.
- Question:Please proof read the text again to eliminate the word repetitions as 'increase' in line 380, extra space in 388, typo TMD instead of TMA in line 405, etc.
Answer: Thanks a lot. The mistakes were revised in our revised manuscript.
- Question:This is the first time I see the term 'thermal durance' - how did you come across it?
Answer: Thanks a lot. It is our spelling mistake. “Thermal durance” should be “Thermal endurance”. We revised the spelling mistakes in our revised manuscript.
- Question:How many times each provided tests were repeated, what is the reproducibility, the percentage of experimental error?
Answer: Thanks a lot. Generally, for the structural characterization tests, such as FTIR, XRD, and so on, only one sample was used. For thermal and optical properties tests, two or three samples were usually used to guarantee the accuracy. For the tensile property tests, 6 samples were used to get the average values of the tensile strength and modulus. For the dielectric constant tests, 6 samples were used to get the average values of the dielectric constants and dissipation factors. The reproducibility was good in our tests and the percentage of experimental error could usually be controlled to be as low as impossible in our tests.
Reviewer 2 Report
Comments and Suggestions for Authors
The authors designed and developed a series of ester-linked polyimide (PEsI) films through copolymerization chemistry. The resulting PEsI films exhibited enhanced blackness, alongside increasing glass transition temperatures (Tg). Overall, this is an interesting and thorough study, but certain improvements are necessary before it can be accepted for publication:
1. Please provide experimental details for the DSC measurements in Section 2.2.
2. If possible, please include the frequency-dependent dielectric spectrum of PEsI films, rather than limiting the results to 10 GHz. For polymer dielectrics, it is important to demonstrate dielectric stability across a range of frequencies.
3. Please check for errors throughout the main text. For example: Line 24: Should this be BDA instead of IDA? Line 349: Should this be Df instead of Dk?
4. How many data points were obtained for measuring dielectric strength? Conducting a Weibull distribution analysis could help solve data variability issues that usually occurs during this kind of experiments.
5. The reported dielectric strength of 150 V/μm seems low for ensuring the "electrical insulating properties" claimed by the authors, especially considering the industrial safety margin (operating electric field typically ~20% of the dielectric strength). Are there targeted working condition parameters or references to support this claim?
6. The authors appear to have used monomers for DFT calculations. Can monomers be considered representative of polymers in quantum chemistry calculations? Alternatively, would using repeating units provide a more accurate analysis?
7. It would be beneficial to calculate the optical bandgap from UV-Vis results and compare it with DFT calculations.
8. Including FTIR spectra of the precursors would be helpful for comparison with the polymers.
9. The reported dielectric constant (Dk) seems high for electronic circuit applications. Please provide supporting evidence or references to confirm whether it complies with industry standards and requirements?
10. The DFT results could be analyzed further. For instance, explain the observed changes in the HOMO levels and relate these to specific electron-donating groups in the polymers.
11. Please include the chemical structures of PAA-I~VI in Figure 2 for better clarity.
12. The details of the different PEsI films in Figure 6 are difficult to observe. Consider separating these into an additional figure for better visibility.
Author Response
- Question: Please provide experimental details for the DSC measurements in Section 2.2.
Answer: Thanks a lot. The experimental details for the DSC measurements were added in our revised manuscript as follows.
“Differential scanning calorimetry (DSC) was carried on a DSC 200 F3 Maia system (NETZSCH, Selb, Germany) at a heating rate of 10 °C/min in nitrogen.”.
- Question: If possible, please include the frequency-dependent dielectric spectrum of PEsI films, rather than limiting the results to 10 GHz. For polymer dielectrics, it is important to demonstrate dielectric stability across a range of frequencies.
Answer: Thanks a lot. This is indeed a good suggestion. However, in our evaluation, the dielectric data at the frequency of 10 GHz is the most concerned parameters. Thus, only the fixture for the 10 GHz test was equipped for the apparatus. In our future work, the frequency-dependent dielectric spectrum of PEsI films might be investigated in detail.
- Question: Please check for errors throughout the main text. For example: Line 24: Should this be BDA instead of IDA? Line 349: Should this be Df instead of Dk?
Answer: Thanks a lot. These are our spelling mistakes. We revised them in our revised manuscript.
- Question: How many data points were obtained for measuring dielectric strength? Conducting a Weibull distribution analysis could help solve data variability issues that usually occurs during this kind of experiments.
Answer: Thanks a lot. In the dielectric strength tests, we measured 8 points in different positions in the PEsI films and the average value was calculated. As for the Weibull distribution analysis, it is really a bit out of present knowledge. It might be evaluated and discussed in our future work.
- Question: The reported dielectric strength of 150 V/μm seems low for ensuring the "electrical insulating properties" claimed by the authors, especially considering the industrial safety margin (operating electric field typically ~20% of the dielectric strength). Are there targeted working condition parameters or references to support this claim?
Answer: Thanks a lot. As suggestion, the expression on the electrical insulating properties of the PEsI films (Nevertheless, the DS values over 150 V/μm could guarantee the electrical insulating properties of the PEsI samples.) was deleted in our revised manuscript.
- Question: The authors appear to have used monomers for DFT calculations. Can monomers be considered representative of polymers in quantum chemistry calculations? Alternatively, would using repeating units provide a more accurate analysis?
Answer: Thanks a lot. We usually use DFT calculations for both of monomers and polymers in our study. For monomers, the lowest unoccupied molecular orbital (LUMO) energy levels (εLUMO) of the dianhydrides, or the highest occupied molecular orbital (HOMO) energy levels (εHOMO) for the diamines were often calculated for the comparison of polymerization activity of the monomers. For polyimides, the energy gaps (Δε) between the HOMO and LUMO energy levels (Δε=|εHOMO-εLUMO|) were often calculated as so to evaluate the difficulty of electron transition. In the current work, the energy gaps (Δε) were simulated and calculated using the repeating unit instead of the monomers.
- Question: It would be beneficial to calculate the optical bandgap from UV-Vis results and compare it with DFT calculations.
Answer: Thanks a lot. This is a good suggestion. The technique for the optical bandgap calculation will be explored in our future work and the results will be compared with the DFT calculations.
- Question: Including FTIR spectra of the precursors would be helpful for comparison with the polymers.
Answer: Thanks a lot. We have tried the comparison of the FTIR spectra of the PAA precursors and the final PEsI films. However, the FTIR spectra of the PAA precursors were severely affected by the residual solvents in the polymers when we precipitate the PAA into ethanol solvent. Thus, only the FTIR spectra of the PEsI films were presented.
- Question: The reported dielectric constant (Dk) seems high for electronic circuit applications. Please provide supporting evidence or references to confirm whether it complies with industry standards and requirements?
Answer: Thanks a lot. The reported dielectric constants (Dk) of the current PEsI films are indeed higher than the requirements. However, the dielectric dissipation factors (Df) seemed to be satisfiable for the industry standards. We are trying to reduce the Dk and maintained the Df values of the PEsI films by introducing new ester-containing monomers, such as the naphthalene-bridged dianhydride. The primary results are promising and the detailed data will be reported in our future work.
- Question: The DFT results could be analyzed further. For instance, explain the observed changes in the HOMO levels and relate these to specific electron-donating groups in the polymers.
Answer: As suggested, the discussion was added in our revised manuscript as follows.
“…In addition, one could observe that the εHOMO values for the NDA-derived polymers (BPTME-NDA and BPDA-NDA) were all higher than those of the ABABP-derived ones (BPTME-ABABP and BPDA-ABABP). This implied the higher electron-donating ability of the NDA components…”.
- Question: Please include the chemical structures of PAA-I~VI in Figure 2 for better clarity.
Answer: As suggestion, the chemical structures of PAA-I~VI was added in Figure 2 in our revised manuscript.
- Question: The details of the different PEsI films in Figure 6 are difficult to observe. Consider separating these into an additional figure for better visibility.
Answer: The inserted PEsI films showed quite similar appearance and thus maintained in Figure 6. If they were treated as an additional figure, the figure might look so monotonous. Thus, it was kept in the current form.